# Enhancement of antiphotoaging properties of *Cannabis sativa* stem water extracts by fermentation with *Lacticaseibacillus casei*

Jin-Woo Kim[1,2☺], Huitae Min[2☺], Jisu Park[2,3], Seongsu Na[2,3], Taejung Kim[2,4], Pahn-Shick Chang[1], Young-Tae Park [2,4*], Jin-Chul Kim[2,4*], Jungyeob Ham[2,4,5*]

1 Interdisciplinary Program in Agricultural Genomics, Seoul National University, Seoul, Republic of Korea, 2 Center for Natural Product Efficacy Optimization, Korea Institute of Science and Technology (KIST), Gangneung, Republic of Korea, 3 Department of Biochemical Engineering, Gangneung-Wonju National University, Gangneung, Republic of Korea, 4 Natural Product Applied Science, KIST School, University of Science and Technology (UST), Gangneung, Republic of Korea, 5 NeoCannBio Co., Ltd., Seoul, Republic of Korea

☺ These authors contributed equally to this study.
* pyt1017@kist.re.kr (YTP); jckim@kist.re.kr (JCK); ham0606@kist.re.kr (JH)

## Abstract

Skin photoaging, driven primarily by UVB radiation, leads to collagen degradation and oxidative stress, contributing to the visible signs of aging, such as wrinkles and loss of skin elasticity. This process is mediated by the upregulation of matrix metalloproteinase-1 (MMP-1), which is triggered by reactive oxygen species, and the activation of photoaging-related signaling pathways, including ERK, JNK, and p65. In the present study, we evaluated the antiphotoaging potential of fermented and non-fermented *Cannabis sativa* stem water extracts, focusing on their ability to suppress MMP-1 expression and reduce oxidative stress in UVB-irradiated human dermal fibroblasts. Unlike previous studies that have primarily focused on leaves or flowers, our study highlights the stems of *C. sativa* as a novel and underutilized source of bioactive compounds for skin protection. Using *Lacticaseibacillus casei* for fermentation, we observed enhanced bioactivity in the fermented extracts, particularly in terms of a 6.6% greater inhibition of MMP-1 expression and 68.3% increased flavonoid content, compared to the non-fermented extracts. Fermented water extract demonstrated the most potent suppression of UVB-induced signaling pathways and collagen breakdown. Our findings suggest that fermentation enhances the antiphotoaging properties of *C. sativa* stems, offering a promising potential for natural, plant-based skin care solutions aimed at preventing UVB-induced skin aging.

## Introduction

The growing awareness of skincare and the increasing demand for natural and effective anti-aging solutions have spurred interest in plant-derived bioactive compounds

**Data availability statement:** All relevant data are within the manuscript and its Supporting Information files.

**Funding:** This study was financially supported by the Technological Innovation R&D Program in the form of a grant (RS-2023-00281400) received by JH. This study was also financially support by the Korea Institute of Science and Technology (KIST) via the Ministry of SMEs and Startups (MSS, Korea) in the form of an intramural grant (2Z07251) received by HM, JP, SN, and Y-TP. This study was also financially support by the Korea Institute of Science and Technology (KIST) via the Ministry of SMEs and Startups (MSS, Korea) in the form of an intramural grant (2Z07256) received by HM, JP, SN, and Y-TP. This study was also financially supported by the Korea Institute of Science and Technology (KIST) via the Ministry of SMEs and Startups (MSS, Korea) in the form of an intramural grant (2E33521) received by J-CK, TK, and J-WK. This study was also financially supported by the Ministry of Science and ICT (MSIT, Korea) in the form of a grant (RS-2021-IN211379) received by TK and Y-TP. This study was also financially supported by the Ministry of Food and Drug Safety, Republic of Korea, in the form of a grant (RS-2024-00332024) received by TK and Y-TP.

**Competing interests:** The authors have declared that no competing interests exist.

[1–3]. Traditional synthetic agents used in skincare often raise concerns about their safety and potential side effects, which has led to a shift towards natural compounds that offer antioxidant, anti-inflammatory, and skin-protective benefits. Antioxidants are of significant interest because of their ability to neutralize reactive oxygen species (ROS), reduce oxidative stress, and inhibit the activation of photoaging-related pathways. These properties make them effective in preventing collagen degradation and combating the visible signs of skin aging. Among the extrinsic factors that contribute to skin aging, chronic exposure to ultraviolet (UV) radiation, specifically UVB, is the primary cause of photoaging [4,5]. UVB exposure induces the production of ROS, which activate signaling pathways involving mitogen-activated protein kinase (MAPK) and nuclear factor-kappa B (p65). These pathways lead to the upregulation of matrix metalloproteinases (MMPs), particularly MMP-1, which is responsible for collagen degradation in the dermis. Collagen breakdown results in wrinkles, the loss of skin elasticity, and other hallmarks of aging. Therefore, targeting ROS and MMP activity has become a central focus in the development of antiphotoaging therapies.

*Cannabis sativa* has gained attention because of its diverse range of bioactive compounds, particularly its leaves and flowers, which are rich in cannabinoids such as cannabidiol (CBD) and tetrahydrocannabinol (THC). [6–8]. These parts have been extensively studied for their anti-inflammatory, antioxidant, and neuroprotective properties in various biomedical applications. However, the stems of *C. sativa* have been largely overlooked, despite containing significant levels of polyphenols and flavonoids, which are known to exhibit strong antioxidant and anti-aging effects. Previous studies have primarily focused on the cannabinoid-driven bioactivity of leaves and flowers, whereas our study explores the non-cannabinoid bioactive potential of *C. sativa* stems. Given that polyphenols and flavonoids are key regulators of oxidative stress and collagen degradation, we hypothesized that *C. sativa* stems could serve as a valuable source for developing natural antiphotoaging agents. Furthermore, this is the first study to evaluate the impact of fermentation on enhancing the bioactivity of *C. sativa* stems, particularly in the context of UVB-induced photoaging. Our results demonstrate that fermentation with Lacticaseibacillus casei significantly increases the flavonoid content, leading to enhanced MMP-1 inhibition and antioxidant activity, thus establishing a novel approach for utilizing C. sativa stems in skincare applications.

Fermentation has emerged as a powerful technique to enhance the bioactivity of plant-derived compounds [9]. Through the metabolic actions of microorganisms, fermentation increases the bioavailability of beneficial compounds, breaks down complex molecules into more potent forms, and produces new bioactive metabolites. Among the various microorganisms used for fermentation, *Lacticaseibacillus casei* (previously known as *Lactobacillus casei*) is a well-studied probiotic strain with a proven ability to enhance the antioxidant and anti-inflammatory properties of plant materials [10,11]. Fermentation with *L. casei* increases the content of flavonoids and polyphenols, two critical compounds known for their antioxidant and antiphotoaging properties. *L. casei* releases glycosidases that break down glycosylated compounds into more bioavailable aglycones, making them more effective at neutralizing ROS and inhibiting the pathways responsible for photoaging. Previous studies have

demonstrated that fermented plant extracts exhibit stronger antioxidant activities and better anti-inflammatory effects than non-fermented extracts, supporting the use of *L. casei* [6,12].

Despite increasing interest in natural compounds for skincare, the potential of *C. sativa* stems remains underexplored, particularly in the context of UVB-induced photoaging [13]. Although the leaves and flowers of *C. sativa* are well known for their cannabinoid content and associated health benefits, the bioactivity of their stems has largely been overlooked [14]. Studies focusing on *C. sativa* stems, specifically their polyphenol and flavonoid contents, are limited, leaving a gap in the understanding of their potential role in skincare and anti-aging applications. Additionally, previous studies have shown that fermentation can increase the antioxidant potential of plant materials [10,15]. However, there is a lack of studies specifically examining the impact of fermentation on *C. sativa* stem extracts, particularly regarding their ability to modulate key markers of photoaging, such as ROS production, MMP-1 expression, and the activation of photoaging-related signaling proteins. This gap presents an opportunity to explore the use of fermented *C. sativa* stems as a novel and natural solution for combating UVB-induced skin damage. Addressing this gap could lead to the development of more effective plant-based skincare products that harness the enhanced bioactivity of fermented extracts.

Recent studies have demonstrated that fermentation with *Lacticaseibacillus spp.* can significantly enhance the antioxidant and anti-inflammatory properties of various plant materials, including oats, whey, and herbal extracts, through the release of phenolic aglycones and organic acids [16–18]. However, to date, no study has investigated this approach using *Cannabis sativa* stems. This makes our work the first to examine how fermentation with *L. casei* can augment the antiphotoaging activity of stem-derived extracts in a UVB-induced skin aging model.

This study addresses a significant gap in current research by investigating the underexplored potential of *C. sativa* stems and their bioactivity in the context of skin aging. This study aimed to introduce a novel approach to enhance the antioxidant and antiphotoaging properties of plant-derived extracts by focusing on the fermentation of *C. sativa* stems with *L. casei*. Given the increasing demand for natural and effective skincare solutions, particularly those that protect against photoaging, the findings of this study have important implications for the development of new plant-based skincare products. Moreover, the findings of this study contribute to a more comprehensive understanding of how fermentation can be used to enhance the bioactivity of plant-derived materials, thereby increasing their efficacy in mitigating oxidative stress and collagen degradation. This study could pave the way for the use of fermented *C. sativa* extracts as valuable ingredients in skincare formulations designed to prevent or reduce the effects of UVB-induced skin aging.

## Materials and methods

### Reagents and chemicals

Gifu anaerobic medium (GAM) was purchased from Kisan Bio (Seoul, Korea). Folin–Ciocalteu reagent, gallic acid, quercetin, trolox, 2,2-diphenyl-1-picrylhydrazyl (DPPH), 2,2′-azino-bis-3-ethylbenzthiazoline-6-sulfonic acid (ABTS), and potassium persulfate were purchased from Sigma-Aldrich (St. Louis, MO, USA). Aluminum (III) chloride was purchased from Tokyo Chemical Industry (Tokyo, Japan). Sodium carbonate was purchased from Showa Chemical Industry Co. Ltd. (Tokyo, Japan).

### Preparation of *C. sativa* stem extracts

The *C. sativa* stems used in this study were obtained from the National Agricultural Cooperative Federation (Andong, Korea). The stems were first dried and finely ground into a powder. To prepare the water extract, 50 g of *C. sativa* stem powder was soaked in 1 L distilled water at room temperature for 72 h. The extraction process was repeated three times to maximize the yield of bioactive compounds. After the extraction process was completed, the water extracts were concentrated using a rotary evaporator under reduced pressure and subsequently freeze-dried to obtain the final powdered extracts.

## Bacterial strain growth control and fermentation

*L. casei* was obtained from the Korean Collection for Type Cultures (Daejeon, Korea) and cultured in GAM broth at 37 °C, and maintained in culture following the manufacturer's instructions. The *C. sativa* extracts were divided into four groups: 0 h (non-fermented *C. sativa* water extract), 48 h (fermented *C. sativa* water extract for 48 h), 72 h (fermented *C. sativa* water extract for 72 h), and 96 h (fermented *C. sativa* water extract for 96 h). *L. casei* culture only (LC Sup) and medium only (GAM) were used as controls. For fermentation, 50 mL of water extracts were combined with 1 mL of *L. casei* culture medium and incubated at 37 °C with continuous shaking at 200 rpm. Following fermentation, the samples were centrifuged at 3000 rpm for 20 min at 4 °C and the supernatants collected.

The mixtures were further concentrated using a rotary evaporator and the final fermented extracts were reconstituted in distilled water at a concentration of 10,000 µg/mL. Before use in subsequent experiments, the extracts were filtered through a 0.22 µm polyvinylidene difluoride (PVDF) membrane to remove any remaining bacterial cells. The fermentation process was designed to enhance the bioactivity of *C. sativa* extracts by increasing the content of flavonoids and other bioactive compounds that are known for their antioxidant and antiphotoaging properties.

## Cell culture and cytotoxicity determination

The Human dermal fibroblast (HDF) cell line was obtained from the American Type Culture Collection (Manassas, VA, USA). All experiments involving human-derived cell lines complied with institutional ethical guidelines and were conducted following the Declaration of Helsinki. As HDFs used in this study were commercially available and obtained from a certified provider, specific ethical approval was not required. HDFs were cultured in Dulbecco's modified Eagle's medium (DMEM) supplemented with 10% fetal bovine serum and 1% penicillin–streptomycin at 37 °C and 5% $CO_2$. Once the cells reached 80–90% confluence, they were detached by incubation with trypsin/EDTA. Cells ($2 \times 10^4$) were seeded in a 96-well plate and incubated overnight at 37 °C and 5% $CO_2$. After removing the medium from each well, the cells were treated with 100 µL of DMEM containing the extracts (0–1000 µg/mL) or control media (LC sup or GAM) and incubated for 24 h without UVB irradiation to assess cytotoxicity. The medium was removed from each well and 100 µL of DMEM containing MTT solution (500 µg/mL) was added and incubated for 1 h. The medium was removed from each well and 100 µL of dimethyl sulfoxide was added to dissolve the formazan crystals. The absorbance was measured at 560 nm using an Infinite M1000 microplate reader (TECAN Ltd., Männedorf, Switzerland).

## MMP-1 enzyme-linked immunosorbent assay (ELISA)

ELISA was performed to confirm downregulation of MMP-1 expression in the UVB-irradiated HDFs. HDFs ($4.5 \times 10^5$) were seeded into 6-well plates and cultivated at 37 °C and 5% $CO_2$ for 24 h. After incubation, the medium was replaced with DMEM containing 1000 µg/mL CW, CE, FCS, FCE, or LC. The other cells were treated with DMEM containing 250 µg/mL ascorbic acid as a positive control. The cells were then incubated for 1 h, irradiated with UVB (25 mJ/cm²) light and cultured at 37 °C and 5% $CO_2$ for 24 h. The supernatant was collected and MMP-1 expression was measured using the MMP-1 DuoSet ELISA kit (R&D Systems, Minneapolis, MN, USA).

## Western blot analysis

HDFs ($4.5 \times 10^5$) were seeded into 6-well plates and cultured at 37 °C and 5% $CO_2$ for 24 h. After incubation, the medium was replaced with DMEM containing LC, FCS, or FCE (1000 µg/mL), or ascorbic acid (250 µg/mL) and incubated for 1 h. After incubation, the cells were exposed to 25 mJ/cm² UVB light and cultured for 24 h. Next, 80 µL of PRO-PREPTM extraction buffer (iNtRON Biotechnology, Seoul, Korea) was added, and the extracts were prepared. The protein lysates were quantified using bovine serum albumin (BSA, Sigma-Aldrich) as a standard using a DCTM Protein Assay Kit (Bio-Rad Laboratories, Hercules, CA, USA). Equivalent amounts (20 µg) of protein were separated by electrophoresis on a

10% sodium dodecyl sulfate-polyacrylamide gel and transferred to a PVDF membrane. The membrane was blocked with 5% skim milk dissolved in Tris-buffered saline containing Tween-20 (TBST) for 1 h at room temperature. The membrane was incubated with the primary antibody diluted in 5% BSA in TBST at 4 °C overnight. The secondary antibody was diluted with 5% BSA in TBST and incubated with the membrane at room temperature for 1 h. Membranes were thoroughly washed with TBST and developed using an enhanced chemiluminescence kit (Advansta, San Jose, CA, USA). The expression of each protein was measured using an iBright CL1000 (Thermo Fisher Scientific, Cleveland, OH, USA). The antibodies used for western blot analysis are listed in Supplementary Table 1.

## ABTS and DPPH ROS scavenging assay

The antioxidant activities of the samples were measured using the ABTS and DPPH scavenging assays. Trolox was used as the standard. Each sample was diluted with 70% ethanol to a final concentration of 10,000 µg/mL. The ABTS radical-scavenging assay was performed as previously described, with minor modification [19]. The ABTS stock solution (7 mM) was mixed with 2.45 mM potassium persulfate and incubated for 14 h at room temperature in the dark. The ABTS stock solution was diluted in ethanol to an absorbance of 0.45 (±0.02) at 734 nm, and 0.9 mL of the diluted ABTS solution was mixed with 0.03 mL of sample and incubated at room temperature for 10 min in the dark. The absorbance of the mixture was measured at 734 nm using a microplate reader. The DPPH radical-scavenging assay was conducted as previously described, with some modifications [20]. Briefly, 150 µL of 0.1 mM DPPH diluted in ethanol was mixed with 50 µL of each sample in a 96-well microplate. DPPH sample mixtures were incubated at room temperature for 30 min in the dark. The absorbance of the mixture was measured at 517 nm using a microplate reader. ABTS and DPPH radical-scavenging activities (%) were calculated using the following equation:

$$\text{Radical} - \text{scavenging activity } (\%) = \{(A_{control} - A_{sample})/A_{control}\} \times 100$$

Where $A_{Control}$ is the absorbance of Trolox and $A_{Sample}$ is the absorbance of the samples.

## Total polyphenol and flavonoid assay

The total polyphenol content of the samples was determined using the Folin–Ciocalteu method [21]. Gallic acid (GA) was used to generate a standard curve. The test sample (0.2 mL, diluted in 70% ethanol) was mixed with 0.2 mL Folin–Ciocalteu reagent and 0.6 mL water. After 5 min, 1 mL of an 8% sodium carbonate water solution and 1 mL of distilled water were added. The reaction was performed at room temperature in the dark for 30 min. After centrifugation, absorbance was measured at 765 nm using a microplate reader. The total polyphenol content was expressed as mg GA equivalents (GAE)/g of sample (mg GAE/5–10 g of sample).

The total flavonoid content was confirmed using the aluminum chloride colorimetric method [22]. Quercetin was used to prepare a standard calibration curve. All test samples were diluted in 70% ethanol and mixed with an equal volume of 2% aluminum chloride. After 60 min at room temperature, absorbance was measured at 420 nm using a microplate reader. The total flavonoid content was expressed as mg of quercetin equivalents (QE)/g of sample (mg QE/5–10 g of sample).

## Measurement of intracellular ROS

Intracellular ROS levels were measured using 2′,7′-dichlorodihydrofluorescein diacetate (DCFH-DA, Sigma-Aldrich), a cell-permeable fluorogenic probe that is deacetylated by intracellular esterases and oxidized by ROS to fluorescent dichlorofluorescein (DCF). HDFs ($2 \times 10^4$ cells/well) were seeded in black 96-well plates and incubated for 24 h. Cells were then treated with 25 mJ/cm² of UVB and incubated for 1 h in serum-free DMEM. After UVB exposure, the cells were treated with 100 µg/mL of the indicated extracts (0 h, 48 h, 72 h, 96 h, LC sup, and GAM) for 24 h. DCFH-DA (10 µM final concentration) was added to each well and incubated at 37 °C for 30 min in the dark. After washing twice with PBS, fluorescence

was measured using a microplate reader (Ex: 485 nm, Em: 535 nm). The ROS levels were normalized to those in the UVB-only group and expressed as relative percentages.

## High-performance liquid chromatography (HPLC) analysis of flavonoids

To quantify quercetin and kaempferol in fermented and non-fermented *Cannabis sativa* stem extracts, high-performance liquid chromatography (HPLC) analysis was performed using a Shimadzu Prominence HPLC system equipped with a photodiode array (PDA) detector. Lyophilized extract powders (100 mg) were dissolved in 10 mL of 70% methanol, vortexed, and sonicated for 30 min. The solution was filtered through a 0.22 μm syringe filter prior to injection. Chromatographic separation was achieved using a C18 reversed-phase column (250 mm × 4.6 mm, 5 μm particle size, Waters) at 30 °C. The mobile phase consisted of solvent A (0.1% formic acid in water) and solvent B (acetonitrile), with the following gradient elution profile: 0–5 min, 15% B; 5–20 min, 15–40% B; 20–25 min, 40–70% B; 25–30 min, 70% B. The flow rate was 1.0 mL/min, and the injection volume was 20 μL. Detection was performed at 370 nm. Standard curves were generated using authentic standards of quercetin and kaempferol (Sigma-Aldrich, ≥ 98% purity), and the results were expressed as μg per gram of extract (μg/g). All measurements were conducted in triplicate.

## Statistical analysis

Statistical analyses were conducted using IBM SPSS (version 24.0; SPSS Inc., Armonk, NY, USA). Differences between multiple groups were analyzed using one-tailed analysis of variance, followed by Tukey's post-hoc test. All experiments were independently conducted in triplicate. Values for each experiment represent the mean of three measurements and are presented as the mean ± standard deviation of three values.

## Results

### Establishment of UVB-induced photoaging cell model

In this study, we established a UVB-induced photoaging cell model to evaluate the antiphotoaging effects of *Cannabis sativa* extracts, as described in the Materials and method section. The model was based on HDF cells, a human fibroblast cell line derived from the dermis, which is commonly used for studying UVB-induced skin damage. The cells were cultured under standard conditions (37°C, 5% $CO_2$), and UVB irradiation was applied when the cells reached 80–90% confluence. The UVB exposure levels ranged from 25 to 70 mJ/cm², which mimics real-life UVB exposure that contributes to skin aging. After UVB irradiation, the cells were incubated for 24 hours to allow the induction of photoaging responses. By establishing this model, we aimed at identifying and validating natural compounds that can prevent or reduce the effects of photoaging.

### Preparation of *C. sativa* stem extract and its fermentation by *L. casei*

The *Cannabis sativa* stem was chosen for this study due to its potential as an underutilized resource with bioactive properties. While the leaves and flowers of *C. sativa* are rich in cannabinoids and flavonoids and have been widely studied for their therapeutic benefits, the stems are primarily used in industrial applications such as textiles and are often overlooked for medicinal purposes. The primary focus of this research was on the antiphotoaging effects of *C. sativa* stem extracts. As described in the Materials and methods section, the *C. sativa* stem extracts were prepared using water as solvents.

### Assessment of cytotoxicity of the fermented and non-fermented extracts

To determine the cytotoxicity of the extracts towards HDFs, an MTT assay was performed (Fig 1). None of the samples, including 0 h, 48 h, 72 h, 96 h, LC sup, and GAM, showed cytotoxicity at the maximum concentration of 1000 μg/mL, maintaining cell viability above 90%. Only the ascorbic acid (AS) control group showed a decrease in viability. Exposure

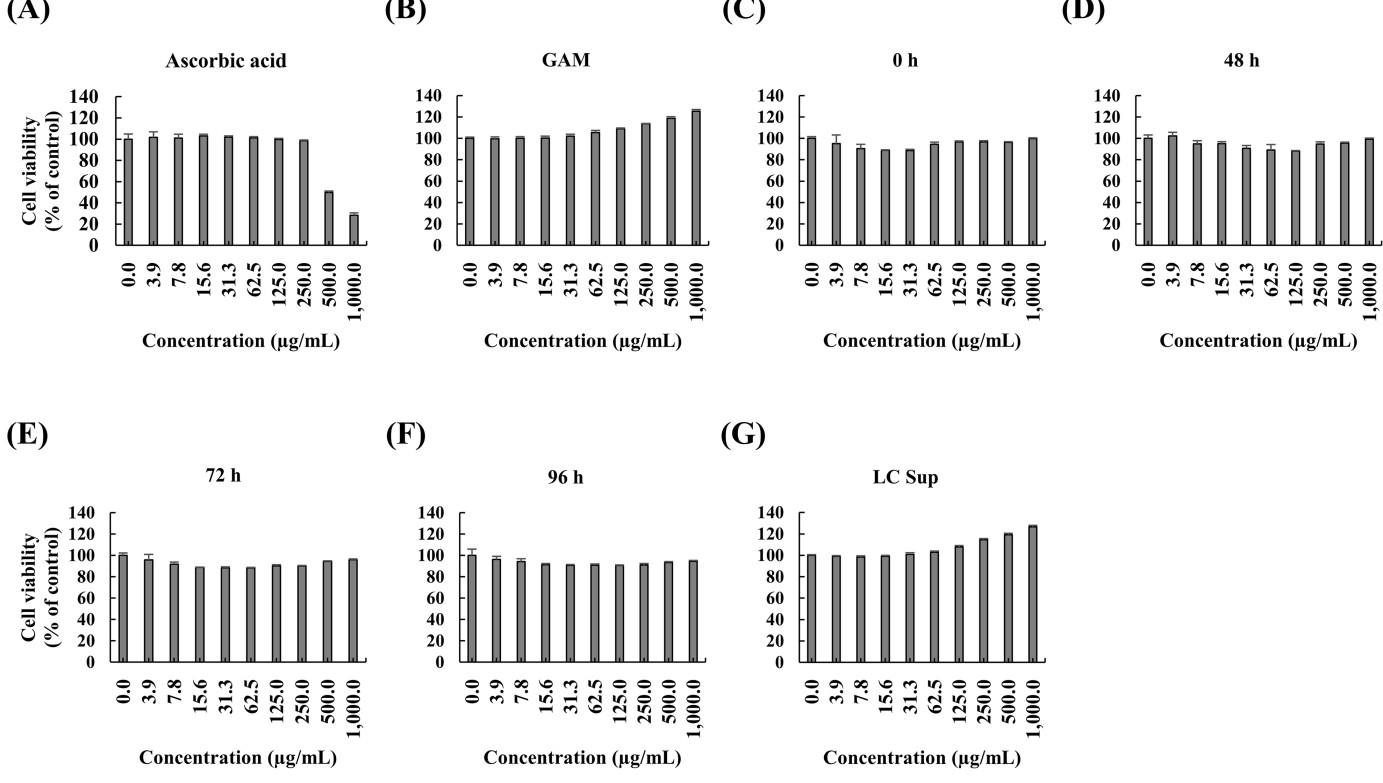

**Fig 1. Cytotoxicity of *Cannabis sativa* stem extracts and their fermented products.** The viability of human dermal fibroblasts was evaluated using an MTT assay. Cells were treated with the indicated various concentrations of the extracts for 24 h, including ascorbic acid, Gifu Anaerobic Medium (GAM), and non-fermented extract (0 h), fermented extracts (48 h and 72 h), and *L. casei* culture supernatant (LC sup). Data are expressed as the mean ± standard deviation (n = 3 per group).

to UVB light resulted in cytotoxicity at a dose of 40 mJ/cm$^2$ or higher (S1 Fig). Therefore, all tested *C. sativa* stem extracts and fermentation-related controls were considered non-toxic and safe for further investigation of their potential antiphotoaging properties.

## Changes in the expression of photoaging-related signaling proteins

UVB irradiation activates several signaling pathways involved in photoaging, primarily through ROS production, which in turn triggers the phosphorylation of key proteins such as ERK, JNK, and p65 [23,24]. These proteins play essential roles in promoting inflammation, oxidative stress, and the degradation of collagen, all of which contribute to skin aging. In particular, the activation of ERK and JNK pathways leads to increased MMP-1 expression, which accelerates collagen breakdown in the dermis. We evaluated the effects of both fermented and non-fermented *C. sativa* extracts on the phosphorylation of ERK, JNK, and p65 in UVB-irradiated HDFs using western blot analysis (Fig 2). UVB exposure markedly increased the phosphorylation levels of ERK, JNK, and p65 compared to non-irradiated controls. Among the treatments, the 72 h fermented extract showed the most prominent inhibitory effects on ERK and p65 phosphorylation, while JNK phosphorylation was most strongly reduced by the 96 h extract, although 48 h and 72 h extracts also showed comparable inhibitory activity. LC sup also exhibited a mild reduction in phosphorylation across all three proteins, suggesting a partial contribution of microbial metabolites. These results indicate that fermented *C. sativa* extracts, particularly those fermented

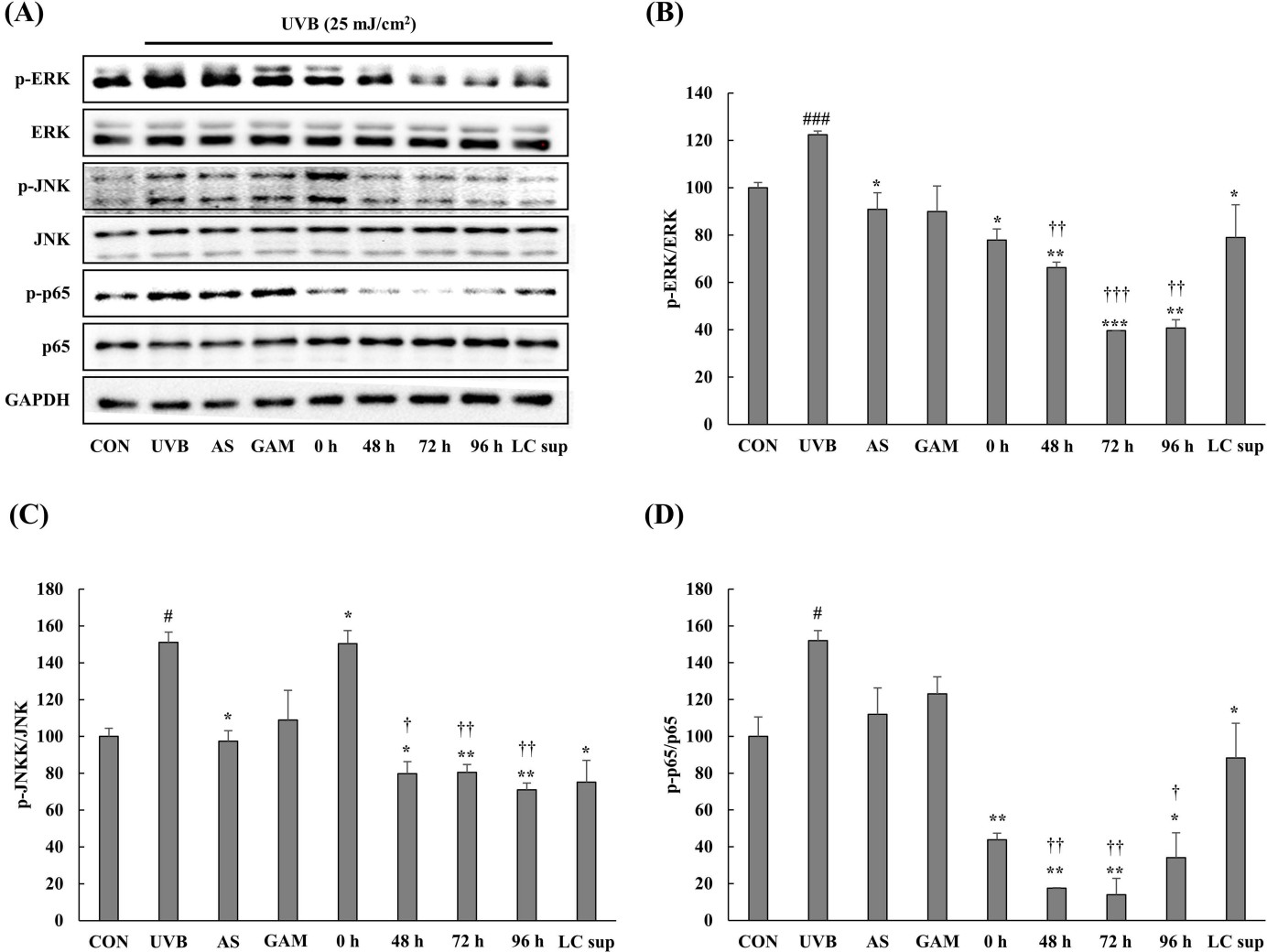

**Fig 2. Effect of the fermented *Cannabis sativa* stem extracts on UVB-induced activation of the signal proteins in human dermal fibroblasts evaluated by western blot analysis.** (A) Representative western blots (n = 1 per lane) for p-ERK, ERK, p-JNK, JNK, p-p65, p65, and GAPDH. (B–D) Western blot densitometry results for (B) p-ERK, (C) p-JNK, and (D) p-p65. Data are expressed as the mean ± standard deviation (n = 3 per group). #$p < 0.05$ and ###$p < 0.001$ compared with CON, ***$p < 0.001$ and *$p < 0.05$ compared with UVB. †$p < 0.05$, ††$p < 0.01$ and †††$< 0.001$ compared with the non-fermented extract (0 **h**). Except for the control group, all other groups were exposed to UVB (25 mJ/cm²).

for 72 h, effectively suppress UVB-induced activation of major pro-photoaging signaling pathways, supporting their potential antiphotoaging efficacy.

## Effect of the fermented extracts on MMP-1 expression

The effects of the extracts on MMP-1 expression were evaluated in UVB-irradiated HDFs using ELISA (Fig 3). As expected, UVB exposure at 25 mJ/cm² significantly elevated MMP-1 levels compared to the non-irradiated control group. AS group exhibited the strongest inhibitory effect on MMP-1 expression, confirming the validity of the assay and the relevance of MMP-1 as a photoaging marker. Among the treatment groups, all fermented extracts (48 h, 72 h, and 96 h) significantly suppressed MMP-1 expression, showing comparable efficacy. These effects were substantially greater than

## ELISA

**(A)** **(B)**

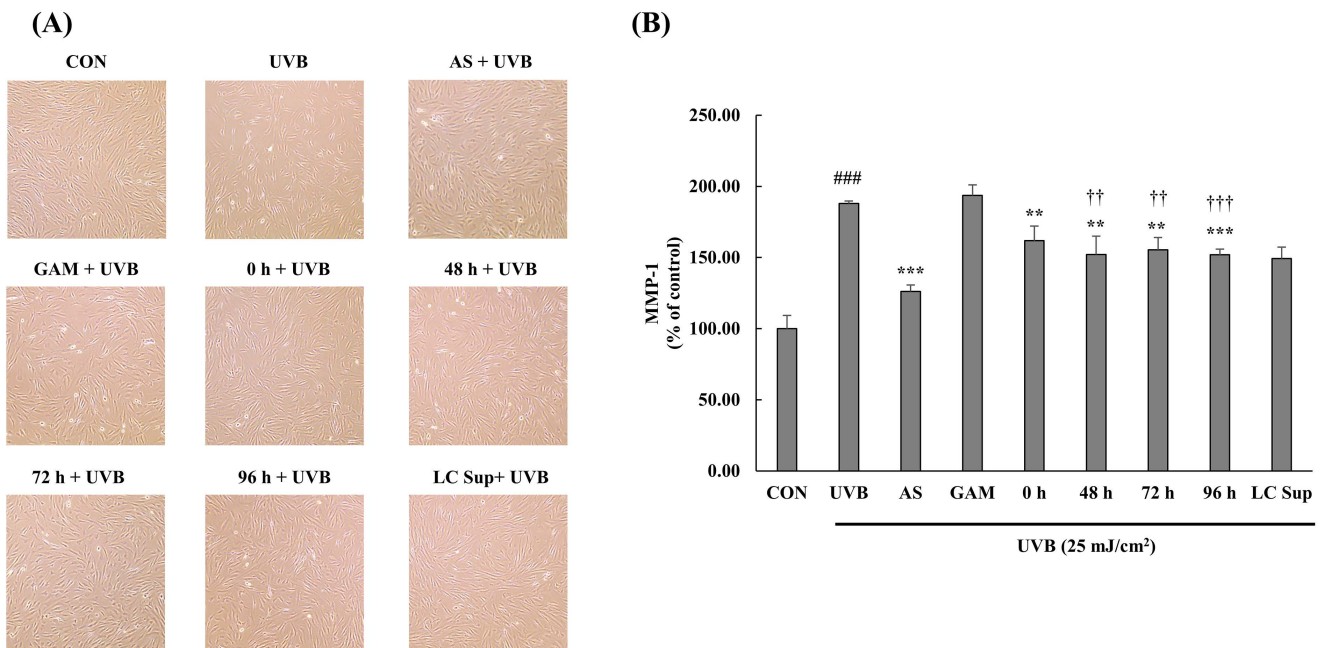

**Fig 3. Fermented *C. sativa* stem extracts attenuate MMP-1 expression induced by UVB irradiation.** (A) Photograph of human dermal fibroblasts and (B) Enzyme-linked immunosorbent assay for MMP-1 expression. All groups except the control were exposed to UVB light (25 mJ/cm²). Data are expressed as the mean ± standard deviation (n = 3 per group). ###$p < 0.001$ compared with CON, ***$p < 0.001$ and **$p < 0.01$ compared with UVB. ††$p < 0.01$ and †††$< 0.001$ compared with the non-fermented extract (0 **h)**.

that of the non-fermented extract (0 h), indicating that fermentation enhanced the anti-photoaging properties of *C. sativa* stems. Interestingly, the LC sup group also led to a statistically significant decrease in MMP-1 levels, suggesting that microbial metabolites may contribute to the observed biological effects. The GAM control did not produce any noticeable change. These findings suggest that fermentation of *Cannabis sativa* stem extracts with *L. casei* enhances their antiphotoaging efficacy, and that both plant-derived and microbial metabolites may contribute synergistically via their fermentation to the protection against UVB-induced skin damage.

### Enhanced antioxidant activity and intracellular ROS suppression by the fermented *C. sativa* stem extracts

ROS generation plays a central role in triggering photoaging through the activation of the MAPK and p65 pathways [25]. To assess the antioxidant effects of the samples and their potential to mitigate this process, ABTS and DPPH radical scavenging activities, total polyphenol content (TPC), total flavonoid content (TFC), and intracellular ROS levels were measured with GAM, 0 h, 48 h, 72 h, 96 h, and LC Sup (Table 1). Fermentation markedly enhanced the antioxidant activity of the extracts. The 72 h sample showed the highest ABTS (63.5 ± 2.4%) and DPPH (52.8 ± 2.1%) radical scavenging activities, followed closely by the 96 h and 48 h extracts. In contrast, the non-fermented extract (0 h) exhibited significantly lower activity (ABTS: 38.4 ± 2.7%, DPPH: 31.2 ± 2.5%), while the GAM and LC Sup controls showed minimal activity (<33%).

Interestingly, TPC decreased slightly after fermentation compared to 0 h (187.8 ± 5.2 mg GAE/g), but remained substantial in all fermented samples (e.g., 153.7 ± 5.1 mg GAE/g at 72 h), suggesting partial polyphenol transformation rather than depletion. This transformation in polyphenol content may be due to the metabolism of certain polyphenolic compounds

**Table 1.  Antioxidant activities, polyphenol and flavonoid contents, and intracellular ROS levels.**

| | GAM | 0 h | 48 h | 72 h | 96 h | LC Sup |
|---|---|---|---|---|---|---|
| **ABTS ROS scavenging activity (%)** | 6.3±0.8 | 38.4±2.7 | 56.1±2.0* | 63.5±2.4** | 61.3±1.9** | 32.7±2.0 |
| **DPPH ROS scavenging activity (%)** | 5.1±0.6 | 31.2±2.5 | 48.5±2.6* | 52.8±2.1** | 50.2±2.3** | 28.3±1.9 |
| **TPC (mg GAE³/g)** | 0.9±0.4 | 187.8±5.2 | 158.2±4.8 | 153.7±5.1 | 151.5±4.9 | 12.5±1.2 |
| **TFC (mg QE⁴/g)** | 1.1±0.2 | 204.8±7.0 | 292.5±5.9* | 387.3±6.9*** | 365.9±6.5** | 5.3±1.8 |
| **Relative ROS level (% of UVB)** | 97.8±2.3 | 88.7±2.1 | 67.1±2.5* | 59.8±2.0** | 61.5±2.4** | 72.6±3.1 |

Asterisks indicate statistically significant differences compared to the 0 h group (*$p < 0.05$, **$p < 0.01$, ***$p < 0.001$).

by *L. casei* during fermentation, as supported by previous studies, which demonstrated that some polyphenols are hydro-lyzed by microflora [26,27]. In contrast, TFC significantly increased with fermentation, peaking at 72 h (387.3±6.9 mg QE/g), representing an approximately 1.9-fold increase over 0 h. To further investigate the effect of fermentation on flavonoid composition, we quantified the levels of two key aglycone flavonoids, quercetin and kaempferol, using HPLC analysis (S3 Fig). In the non-fermented extract (0 h), quercetin and kaempferol were detected at 3.2±0.4 µg/g and 2.1±0.3 µg/g extract, respectively. These levels increased progressively with fermentation time, reaching peak values at 72 h (quercetin: 11.3±0.7 µg/g; kaempferol: 9.0±0.5 µg/g), representing approximately 3.5-fold and 4.3-fold increases, respectively. A slight decline was observed at 96 h, though levels remained elevated compared to 0 h. As expected, no detectable levels of either flavonoid were found in GAM or LC Sup, confirming that the observed flavonoids were plant-derived and not microbial byproducts. These results suggest that fermentation with *L. casei* enhances the release or transformation of flavonoid aglycones in *C. sativa* stem extracts, possibly via enzymatic hydrolysis of glycosylated precursors.

Intracellular ROS levels, measured by DCFH-DA assay in UVB-irradiated HDFs, were significantly reduced by all fermented extracts. The 72 h sample again demonstrated the strongest effect, lowering ROS to 59.8±2.0% of the UVB-only group, compared to 88.7±2.1% in the 0 h group. LC sup also showed a moderate ROS-suppressive effect (72.6±3.1%), while GAM had no significant impact.

These results suggest that fermentation with *L. casei* enhances the antioxidant potential of *C. sativa* stem extracts by increasing flavonoid content and improving intracellular ROS suppression, with optimal effects observed at 72 h.

## Discussion

In this study, we evaluated the antiphotoaging effects of fermented and non-fermented *C. sativa* stem water extracts in UVB-irradiated human dermal fibroblasts. Fermentation with *L. casei* significantly enhanced the biological activity of the extracts, as evidenced by increased flavonoid content and antioxidant capacity. The fermented samples, particularly the 72 h extract, markedly suppressed MMP-1 expression compared to the non-fermented control (0 h), thereby mitigating collagen degradation (Fig 3). Additionally, intracellular ROS levels were significantly reduced, which aligned with the elevated ABTS and DPPH scavenging activities and HPLC-quantified increases in quercetin and kaempferol (Table 1, S3 Fig). Furthermore, the fermented extracts downregulated the UVB-induced phosphorylation of ERK, JNK, and p65, which are key mediators of inflammation and matrix degradation in photoaging (Fig 2). These findings suggest that microbial fermentation enhances the bioavailability and efficacy of *C. sativa* stem extracts, supporting their potential as safe and effective natural antiphotoaging agents.

These findings are consistent with those of previous studies on the antioxidant and anti-inflammatory properties of *C. sativa* [14,28]. However, this study uniquely emphasizes the potential of *C. sativa* stems, particularly when fermented. Incorporating *C. sativa* leaves and flowers as comparative baselines was considered; however, experimental use of these

plant parts requires additional regulatory approvals due to their higher cannabinoid content. Many jurisdictions impose restrictions on handling and researching cannabinoid-rich materials, necessitating specific permits for experimental use. Since this study aimed to evaluate whether fermentation enhances the bioactivity of stems independent of cannabinoids, we prioritized a more accessible and regulatory-compliant approach. While most previous studies have focused on the leaves and flowers, this study shows that fermentation enhances the bioactivity of stems, especially by reducing MMP-1 expression and mitigating UVB-induced photoaging. Fermentation is known to improve the bioavailability of plant compounds, and in this study, fermentation significantly enhanced the suppression of the ERK, JNK, and p65 pathways, which are key regulators of photoaging. Compared to non-fermented extracts, fermented extracts showed superior efficacy in reducing collagen degradation, offering new insights into the potential use of fermented *C. sativa* in natural skincare products.

The fermented samples treatment effectively inhibited phosphorylation of ERK, JNK, and p65, leading to reduced MMP-1 expression and collagen degradation. Fermentation plays a crucial role in enhancing their bioactivity by increasing its flavonoid content, which, in turn, enhances its antioxidant properties (Table 1). These antioxidants neutralize ROS, reducing oxidative stress, and preventing ERK, JNK, and p65 activation. By simultaneously lowering ROS levels and directly inhibiting molecular pathways associated with photoaging, they provide a comprehensive mechanism to protect against UVB-induced skin damage, making it a highly promising natural agent for anti-aging applications, with fermentation significantly enhancing its therapeutic potential.

Although our study demonstrated significant antiphotoaging effects of fermented *C. sativa* stem extracts in UVB-irradiated HDFs, further validation in in vivo models is required to confirm these findings under physiologically relevant conditions. Murine photoaging models, particularly hairless mice (HR-1) or SKH-1 mice, which have been widely used to study UV-induced skin damage, would be appropriate for evaluating the protective effects of fermented extracts against photoaging. These models enable the assessment of collagen degradation, epidermal thickness, and oxidative stress markers in response to chronic UV exposure. Future studies will focus on evaluating the efficacy of fermented *C. sativa* extracts in these preclinical *in vivo* models to establish their potential for topical skin care applications

Our study primarily focused on UVB-induced photoaging; however, UVA radiation and environmental pollutants are also significant contributors to skin aging through oxidative stress and chronic inflammation. Unlike UVB, which mainly affects the epidermis, UVA penetrates deeper into the dermis, leading to collagen degradation and elastin fiber breakdown via ROS generation [29]. Moreover, environmental stressors such as airborne pollutants, particulate matter (PM2.5), and polycyclic aromatic hydrocarbons (PAHs) are known to induce oxidative stress, contributing to premature skin aging and inflammatory responses [30,31]. Given that fermented C. sativa stem extracts exhibit potent antioxidant and anti-inflammatory properties, they may also provide protective effects against UVA-induced photoaging and pollution-related skin damage. Future studies will explore the efficacy of these extracts in mitigating UVA-induced oxidative stress and pollutant-triggered skin inflammation in both in vitro and in vivo models to broaden their potential applications in skin protection.

Although we performed targeted HPLC analysis to quantify specific flavonoids such as quercetin and kaempferol, we acknowledge that this does not fully capture the complexity of fermentation-derived bioactive changes. To more comprehensively elucidate the mechanism underlying the enhanced bioactivity, future studies will incorporate untargeted metabolomic analyses, such as LC-MS, to identify and characterize additional metabolites produced during fermentation that may contribute to the observed antioxidant and antiphotoaging effects. Although more research is required to fully elucidate the mechanisms and broaden its applications, the fermented samples shows great promise as a natural agent for combating UVB-induced skin aging and could play a key role in the development of effective plant-based anti-aging solutions.

## Conclusions

This study demonstrated that the fermentation of *C. sativa* stem water extracts with *L. casei* enhanced their antioxidant and antiphotoaging properties. The fermented water extracts showed significant potential for attenuating UVB-induced

oxidative stress, suppressing MMP-1 expression, and inhibiting key signaling pathways involved in photoaging. These findings suggest that fermented *C. sativa* extracts are promising natural agents for preventing UVB-induced skin aging and promoting skin health.

## Supporting information

**S1 Data.  Raw data for Fig 1 (MTT-based cell viability assay).** This dataset includes raw absorbance-derived cell viability percentages of human dermal fibroblasts (HDFs) treated with various concentrations (0, 100, 250, 500, and 1000 μg/mL) of *Cannabis sativa* stem extracts, non-fermented (0 h) and fermented (48 h, 72 h, 96 h), as well as *Lacticaseibacillus casei* culture supernatant (LC Sup), Gifu Anaerobic Medium (GAM), and ascorbic acid. Viability was assessed by MTT assay after 24 h treatment without UVB exposure. Data are presented as individual replicate values (n = 3) and corresponding mean ± standard deviation for each condition.
(PDF)

**S2 Data.  Raw data for Supplementary Fig 1 (MTT assay: UVB cytotoxicity).** This dataset presents the raw absorbance values of HDFs exposed to increasing doses of UVB light (0–70 mJ/cm²) for 24 hours.
(PDF)

**S3 Data.  Raw data for Fig 3B (MMP-1 ELISA assay).** This dataset provides the calculated MMP-1 concentrations (ng/mL) in HDFs treated with non-fermented (0 h) and fermented *C. sativa* stem extracts (48 h, 72 h, 96 h), LC Sup, GAM, and ascorbic acid (positive control), following UVB irradiation at 25 mJ/cm². All treatment groups except the untreated control were UVB-irradiated. MMP-1 levels in culture supernatants were measured 24 hours after treatment. Each condition was tested in triplicate (n = 3), and data are reported as mean ± standard deviation in ng/mL.
(PDF)

**S4 Data.  Raw data for Supplementary Fig 3 (Quantification of quercetin and kaempferol).** This dataset contains the quantified concentrations (μg/g extract) of quercetin and kaempferol in *C. sativa* stem extracts fermented for 0 h, 48 h, 72 h, and 96 h. GAM medium and LC Sup were also analyzed. Quantification was performed by HPLC.
(PDF)

**S1 Table.  Antibodies used for western blot analysis.** 1, Phospho-SAPK/JNK (Thr183/Tyr185) (G9) Mouse mAb; 2, SAPK/JNK Antibody; 3, Phospho-p44/42 MAPK (Erk1/2) (Thr202/Tyr204) (D13.14.4E) XP® Rabbit mAb; 4, p44/42 MAPK (Erk1/2) (137F5) Rabbit mAb; 5, Phospho-NF-κB p65 (Ser536) (93H1) Rabbit mAb; 6, NF-κB p65 (C22B4) Rabbit mAb; 7, GAPDH (14C10) Rabbit mAb; 8, mouse anti-rabbit IgG-HRP; 9, m-IgGκ BP-HRP.
(PDF)

**S1 Fig.  Assessment of cytotoxicity of the exposure to UVB light.** The viability of human dermal fibroblasts was evaluated using an MTT assay. Cells were irradiated with UVB (0–70 mJ/cm²) for 24 h. Data are expressed as the mean ± standard deviation (n = 3 per group).
(PDF)

**S2 Fig.  Western blot analysis of signaling pathways involved in UVB-induced photoaging.** HDFs were exposed to UVB (25 mJ/cm²) and treated with *C. sativa* stem water extracts (0 h, 48 h, 72 h, 96 h), LC Sup, or GAM for 24 h. Protein expression levels were assessed using antibodies against phosphorylated and total forms of ERK, JNK, and p65. GAPDH was used as a loading control. Representative blot images are shown for (A) p-ERK, (B) total ERK, (C) p-JNK, (D) total JNK, (E) p-p65, (F) p65, and (G) GAPDH. Molecular weights are indicated in kilodaltons (kDa).
(PDF)

**S3 Fig. Quantification of quercetin and kaempferol.** The concentrations of quercetin and kaempferol were determined in C. sativa stem extracts fermented for 0 h, 48 h, 72 h, and 96 h, as well as in controls including GAM, LC Sup, and the non-fermented 0 h sample.
(PDF)

## Author contributions

**Conceptualization:** YOUNG-TAE PARK, Huitae Min, Jin-Chul Kim, Jungyeob Ham.

**Data curation:** Jin-Woo Kim, Huitae Min.

**Formal analysis:** Jin-Woo Kim.

**Methodology:** Huitae Min, Pahn-Shick Chang.

**Project administration:** Huitae Min.

**Software:** Jisu Park, Taejung Kim.

**Supervision:** Jin-Chul Kim, Jungyeob Ham.

**Validation:** Seongsu Na.

**Writing – original draft:** Jin-Woo Kim, Huitae Min.

**Writing – review & editing:** Jin-Woo Kim, Huitae Min.

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
