## [Decision Letter · Decision Letter 0]

3 Jan 2025

Dear Dr. PARK,

Thank you for submitting your manuscript to PLOS ONE. After careful consideration, we feel that it has merit but does not fully meet PLOS ONE’s publication criteria as it currently stands. Therefore, we invite you to submit a revised version of the manuscript that addresses the points raised during the review process.

We look forward to receiving your revised manuscript.

Kind regards,

Taher Hatahet, Ph.D

Academic Editor

PLOS ONE

Journal Requirements:

Additional Editor Comments:

Please address the comments form reviewer one and two.

Regarding blot/gel data: PLOS ONE now requires that submissions reporting blots or gels include original, uncropped blot/gel image data as a supplement or in a public repository. This is in addition to complying with our image preparation guidelines described at https://journals.plos.org/plosone/s/figures#loc-blot-and-gel-reporting-requirements. These requirements apply both to the main figures and to cropped blot/gel images included in Supporting Information.

Reviewers' comments:

Reviewer's Responses to Questions

**Comments to the Author**

1. Is the manuscript technically sound, and do the data support the conclusions?

Reviewer #1: Yes

Reviewer #2: Partly

Reviewer #3: Yes

2. Has the statistical analysis been performed appropriately and rigorously?

Reviewer #1: Yes

Reviewer #2: N/A

Reviewer #3: Yes

3. Have the authors made all data underlying the findings in their manuscript fully available?

Reviewer #1: Yes

Reviewer #2: Yes

Reviewer #3: Yes

4. Is the manuscript presented in an intelligible fashion and written in standard English?

Reviewer #1: No

Reviewer #2: Yes

Reviewer #3: Yes

Reviewer #1: Below is a structured review of the manuscript titled "Enhancement of antiphotoaging properties of Cannabis sativa stem extracts by fermentation with Lacticaseibacillus casei". Comments are categorized into major and minor points based on their criticality for improving the manuscript.

Major Comments

Abstract and Objectives Clarity:

The abstract concisely summarizes the study but lacks specific numerical data or comparative metrics to substantiate claims (e.g., "enhanced bioactivity"). Including key results such as specific percentage improvements in antioxidant activity or reductions in MMP-1 expression would strengthen the abstract.

The study's objectives in the Introduction section should explicitly mention the novelty of using Cannabis sativa stems, differentiating it from prior work focusing on other plant parts.

Experimental Design:

The choice of fermentation duration (72 hours) and conditions (e.g., shaking speed, temperature) for L. casei lacks justification. Why were these parameters chosen, and were any preliminary optimization experiments conducted?

Control comparisons: The inclusion of ascorbic acid as a positive control is appropriate, but the authors should also compare fermented and non-fermented extracts statistically to determine the significance of fermentation specifically.

Data Analysis and Representation:

The manuscript does not report effect sizes or confidence intervals for the data, which are critical for assessing biological relevance, especially in translational contexts.

Figures lack sufficient annotations to guide interpretation. For example, Figure 2 (Western blot) should indicate molecular weight markers and justify the specific signaling pathways selected.

Biological Relevance and Clinical Translation:

The in vitro model using HDFs is appropriate but limited. The discussion acknowledges this but could be improved by suggesting specific in vivo models (e.g., murine photoaging models) for future validation.

The focus solely on UVB-induced damage ignores the potential broader applicability of the extracts, such as protection against UVA or environmental pollutants.

Mechanistic Insights:

The discussion on fermentation enhancing bioactivity (e.g., flavonoid content) is convincing but speculative. Are there any specific fermentation byproducts identified that correlate with observed effects (e.g., HPLC or LC-MS data)? Without such analysis, the mechanistic claims are incomplete.

Ethical Considerations:

While the manuscript states that no ethical approval was required, the use of human-derived dermal fibroblasts demands a clearer mention of the cell line's origin and adherence to ethical standards for cell line use.

Minor Comments

Language and Terminology:

Certain phrases, such as "fermentation significantly enhanced bioactivity," are vague. Replace with specific metrics (e.g., "fermentation increased flavonoid content by X%, enhancing antioxidant activity by Y%").

Repetition in the Introduction regarding Cannabis sativa’s underexplored potential in skincare can be condensed.

Figures and Tables:

Table 1 should include significance levels for each comparison, not just different letter labels. Use asterisks or p-values for clarity.

Figures should have high resolution in the final submission to avoid interpretive issues during review.

Citations and Literature Integration:

Several references are over five years old. Consider citing recent studies on plant-based antiphotoaging agents or Cannabis sativa bioactivity.

The introduction lacks a detailed comparison with other plant extracts fermented with Lacticaseibacillus spp. Addressing this would position the study better in the context of current literature.

Formatting Issues:

Ensure uniformity in referencing styles, particularly with et al. usage.

Standardize units (e.g., μg/mL vs mg/mL) across the manuscript for consistency.

Limitations Section:

While some limitations are mentioned, the manuscript should specifically discuss the absence of mechanistic validations (e.g., metabolomic analyses of fermentation products) and the lack of UVA or other stressor evaluation.

Supplementary Information:

The supplementary data should detail the antibodies used in Western blotting, including catalog numbers and suppliers, for reproducibility.

Overall Assessment

The manuscript addresses a novel area of research with potential applications in natural skincare. However, it requires substantial revisions to improve clarity, depth, and rigor. Addressing the major comments will ensure the study's findings are robust and impactful, while minor comments focus on improving readability and presentation.

Reviewer #2: The study investigates the antiphotoaging properties of fermented Cannabis sativa stem extracts using a human dermal fibroblast (HDF) model. While the authors emphasize the underexplored biological effects of cannabis stems, they also acknowledge prior research on cannabis leaves and flowers. Although stems and leaves differ significantly in their bioactive substance content, the claimed novelty of this study warrants careful consideration.

The manuscript has several strengths, including its focus on an underutilized part of the plant and its exploration of fermentation as a means of enhancing bioactivity. However, there are several critical issues and clarifying points that should be addressed before the study can be considered for publication.

Key Comments and Questions

1. In the section describing cytotoxicity determination, it is unclear whether all samples (including those with extracts) were UVB-irradiated. If so, please clarify the rationale behind this approach. Cytotoxicity assay should only evaluate one compound/extract at a time.

2. On line 155, the term "CW" appears but is not defined in the methods section. If it refers to the cannabis water extract, why is it labeled "CW" here instead of "CS," as initially described?

3. The LC control group, which is defined as the culture supernatant of Lacticaseibacillus casei, shows notable biological effects in several assays, including:

- MMP-1 ELISA

- ROS scavenging assays (ABTS and DPPH)

- Polyphenol and flavonoid content assays

- How do the authors interpret these results? Given that LC is a control group without plant extracts, its high activity raises concerns about the contributions of bacterial metabolites or medium components.

4. While the focus on stems is unique, the inclusion of cannabis leaves and flowers in the study could have provided a comparative baseline. Were these parts excluded due to practical constraints or specific research objectives?

5. The manuscript mentions UVB-induced activation of ERK, JNK, and p65 as part of the photoaging model. However, it is unclear whether these signaling changes were the sole basis for evaluating photoaging. Were additional endpoints (e.g., collagen degradation or ROS production) assessed?

The activation of JNK in the UVB group appears less convincing. Could the authors provide a clearer explanation or replicate the data to confirm this pathway's role in UVB-induced photoaging?

Additionally, in the FCS group, phosphorylated p65 (p-p65) appears absent, whereas the CON group shows a baseline level of p-p65. This raises questions about the reproducibility of the data or the experimental conditions. Could the authors explain why p-p65 is entirely suppressed in FCS and not in the control?

6. The study reports a nearly two-fold increase in total flavonoid content in the fermented extract (FCS) compared to the non-fermented extract (CS). While fermentation is said to enhance flavonoid production, the specific mechanism remains unexplained. Do the authors attribute this to microbial enzymatic activity, precursor transformation, or other factors?

7. On line 322, the manuscript claims that FCS reduced UVB-induced ROS levels. However, no assay directly measuring ROS levels in the presence of UVB appears in the paper. Can the authors clarify or modify this statement?

8. - The ABTS results are described as significantly different between groups, but the data show minimal variation (e.g., 88.6% for FCS vs. 89.5% for CS). This should be addressed.

- In the DPPH assay, the LC control group demonstrated the highest scavenging potential, even outperforming the fermented extract (FCS). This observation contradicts the claim that fermentation enhances scavenging activity. How do the authors explain this?

9. -The LC group is defined as “L. casei culture without plant extracts”, yet it shows measurable levels of flavonoids and polyphenols. This finding is puzzling. Could this be due to assay interference, microbial metabolites, or contamination? Additional clarification is essential.

Summary

The study offers an intriguing approach to enhancing the antiphotoaging properties of Cannabis sativa stems through fermentation. However, several methodological inconsistencies and data interpretation issues need to be resolved. Specifically:

- Clarification of UVB irradiation protocols.

- Addressing discrepancies in the LC group’s activity.

- Providing mechanistic insights into the effects of fermentation on flavonoid and polyphenol levels.

- Revisiting claims related to ROS assays and ABTS/DPPH results.

Once these issues are addressed, the study will provide a stronger contribution to the field of natural anti-photoaging agents.

Reviewer #3: The authors have presented a study on the enhancement of anti-photo-aging properties of Cannabis sativa stem extracts by fermentation. The manuscript is well written, extremely detailed and easy to understand.

**Do you want your identity to be public for this peer review?** For information about this choice, including consent withdrawal, please see our Privacy Policy

Reviewer #1: No

Reviewer #2: No

Reviewer #3: No

---

## [Author Response · Author response to Decision Letter 1]

17 Jul 2025

Dear Dr. Hatahet,

We sincerely thank you and the reviewers for the thoughtful and constructive comments on our manuscript. We have carefully revised the manuscript to address all major and minor concerns. The revised version incorporates the following key changes:

1. Cytotoxicity assay clarification

2. New ROS measurement experiment: We conducted an additional intracellular ROS assay using DCFH-DA under UVB conditions.

3. Western blot replication and clarification: We repeated the Western blot experiments, confirming reproducible activation of ERK, JNK, and p65 upon UVB exposure. The 72 h fermented extract (FCS) strongly suppressed phosphorylation of these markers, as supported by densitometric analysis.

4. Mechanistic insight enhanced: We added a mechanistic explanation for the increase in flavonoid content, attributing it to enzymatic hydrolysis by L. casei. HPLC data confirm elevated levels of quercetin and kaempferol after fermentation.

5. Updated antioxidant assays and controls: We repeated both ABTS and DPPH assays using freshly prepared samples and included additional control groups (GAM, LC Sup, 0 h). Statistical analysis using asterisk-based notation was applied. The revised results better support the interpretation that fermentation enhances antioxidant activity.

6. Clarification on LC group activity: We addressed the unexpected activity in the LC Sup group by including a GAM-only control and performing HPLC. No flavonoids were detected in LC Sup, and the results are now interpreted as due to microbial metabolites, not contamination or media interference.

7. Formatting and transparency improvements:

o Table 1 updated with statistical significance indicators.

o Molecular weight markers added to all Western blots.

8. Expanded Discussion and limitations:

o Inclusion of murine models (HR-1, SKH-1) for future in vivo validation.

o Broader applicability to UVA and pollutant-induced skin damage discussed.

o Acknowledgement of the need for future untargeted metabolomic profiling.

We believe that these revisions have significantly strengthened the manuscript in both scientific rigor and clarity. A detailed point-by-point response to each reviewer comment is included in the revised submission.

Thank you again for your time and consideration.

Sincerely,

Young-Tae Park, Ph.D.

Center for Natural Product Efficacy Optimization, KIST, Gangneung, 25451, Republic of Korea

pyt1017@kist.re.kr

---

## [Editor Report · Decision Letter 1]

21 Jul 2025

Enhancement of antiphotoaging properties of Cannabis sativa stem water extracts by fermentation with Lacticaseibacillus casei

PONE-D-24-50437R1

Dear Dr. PARK,

We’re pleased to inform you that your manuscript has been judged scientifically suitable for publication and will be formally accepted for publication once it meets all outstanding technical requirements.

Kind regards,

Taher Hatahet, Ph.D

Academic Editor

PLOS ONE

Additional Editor Comments (optional):

Dear Author

thanks for addressing reveiwers comments

Kind regards
---

## [Editor Report · Acceptance letter]

PONE-D-24-50437R1

PLOS ONE

Dear Dr. PARK,

I'm pleased to inform you that your manuscript has been deemed suitable for publication in PLOS ONE. Congratulations! Your manuscript is now being handed over to our production team.

Kind regards,

on behalf of

Dr. Taher Hatahet

Academic Editor

PLOS ONE